# First-Trimester Fetal Hepatic Artery Examination for Adverse Outcome Prediction

**DOI:** 10.3390/jcm11082095

**Published:** 2022-04-08

**Authors:** Bartosz Czuba, Piotr Tousty, Wojciech Cnota, Dariusz Borowski, Agnieszka Jagielska, Mariusz Dubiel, Anna Fuchs, Magda Fraszczyk-Tousty, Sylwia Dzidek, Anna Kajdy, Grzegorz Świercz, Sebastian Kwiatkowski

**Affiliations:** 1Department of Obstetrics and Gynecology, Medical University of Silesia, 41-703 Ruda Slaska, Poland; bartosz.czuba@sonomed.net (B.C.); woytek@eth.pl (W.C.); majonez7@wp.pl (A.J.); 2Department of Gynecology and Obstetrics, Pomeranian Medical University, 70-111 Szczecin, Poland; sylwiadzidek@wp.pl (S.D.); kwiatkowskiseba@gmail.com (S.K.); 3Department of Perinatology, Gynecology and Gynecologic Oncology, Collegium Medicum, Nicolaus Copernicus University, 85-821 Bydgoszcz, Poland; darekborowski@gmail.com; 4Department of Obstetrics, Gynecology and Gynecological Oncology, Jan Biziel University Hospital, Collegium Medicum, Nicolaus Copernicus University, 85-168 Bydgoszcz, Poland; profdubiel@gmail.com; 5Chair and Department of Gynecology, Obstetrics and Oncological Gynecology, Medical University of Silesia in Katowice, 40-211 Katowice, Poland; afuchs999@gmail.com; 6Department of Neonatal Diseases, Pomeranian Medical University, 70-111 Szczecin, Poland; magfraszczyk@gmail.com; 7Department of Reproductive Health, Centre of Postgraduate Medical Education, 01-004 Warsaw, Poland; anna.kajdy@cmkp.edu.pl; 8Clinic of Obstetrics and Gynecology, Provincial Combined Hospital in Kielce, 25-736 Kielce, Poland; swierczag@poczta.onet.pl; 9Collegium Medicum, Jan Kochanowski University in Kielce, 25-369 Kielce, Poland

**Keywords:** adverse outcome, hepatic artery, chromosomal abnormalities, congenital heart defects

## Abstract

Objective: To assess whether there are differences in first-trimester fetal hepatic artery flows depending on pregnancy outcomes. Methods: The prospective study conducted in 2012–2020 included 1841 fetuses from singleton pregnancies assessed during the routine first-trimester ultrasound examination (between 11- and 14-weeks’ gestation). Also, each fetus was examined to determine their hepatic artery flows by measuring the artery’s pulsatility index (HA-PI) and peak systolic velocity (HA-PSV). Results: The fetuses that were classified as belonging to the adverse pregnancy outcome group (those with karyotype abnormalities and congenital heart defects) were characterized by a significantly lower HA-PI and higher HA-PSV compared to normal outcome fetuses. Conclusion: Hepatic artery flow assessment proved to be a very useful tool in predicting adverse pregnancy outcomes, in particular karyotype abnormalities and congenital heart defects.

## 1. Introduction

Recent years have seen a continuous dynamic development in prenatal diagnosis and the discovery of increasingly improved tools for detecting congenital fetal defects, aneuploidy, or pregnant patients at risk of developing different forms of placental insufficiency [1,2,3]. The inversion of the pyramid of antenatal care and the focus on first-trimester screening have led to significantly improved perinatal outcomes [4]. Nowadays, no one could imagine first-semester diagnosis without the ultrasound evaluation of such parameters as nuchal translucency (NT), fetal heart rate (FHR), ductus venosus pulsatility index (DV-PI), or uterine artery pulsatility index (Ut-PI), which together with the biochemical factors, such as the β-subunit of hCG gonadotropin (beta-hCG), pregnancy-associated plasma protein A (PAPP-A), and placental growth factor (PlGF), are well-researched prognostic factors for pregnancy outcomes [1,2,3]. However, a continued search for new markers may contribute to improving the detection sensitivity for various abnormalities.

The liver is one of the most important organs during fetal life, where it has a hematopoietic function and regulates the entire metabolism of the fetus. Physiologically, most of the vascularity of the liver during fetal life derives from the umbilical vein and the portal vein, and only about 10% from the hepatic artery [5]. It has been proven that in the event of hypoxemia and fetal growth disorders in the second and third trimesters of pregnancy, this proportion is disturbed [6,7]. The flow through the liver coming from the veins–becoming more and more reduced under these conditions–leads to dilation of the hepatic artery related to a local increase in adenosine concentration aimed at compensating for this abnormal flow. This mechanism is known as HABR (hepatic arterial buffer response) [8].

The few reports of studies on small groups available indicate that hepatic artery flow assessment may be a predictive factor for the occurrence during pregnancy of chromosomal disorders and other adverse pregnancy outcomes [9,10].

The first aim of the study was to use ultrasound to assess the hepatic artery flow in the first trimester as a predictive factor for the occurrence of pregnancy adverse outcomes. The second aim was to identify the flow in the same artery depending on the type of karyotype abnormalities detected during pregnancy or after delivery or miscarriage. 

## 2. Patients and Methods

The prospective study conducted in the years 2012–2020 at Sonomedico Żory and in the Clinical Department of Obstetrics and Gynecology in Ruda Śląska included 1841 fetuses from healthy singleton pregnancies during the routine first-trimester ultrasound examination (between 11- and 14-weeks’ gestation) for fetal defects and the risk of aneuploidy. The patients qualified for the study had a negative history of adverse pregnancy outcomes and had no concurrent diseases. The ultrasound was performed in accordance with the Fetal Medicine Foundation (FMF) principles for first-trimester pregnancies to evaluate the anatomy of the fetus and take the following measurements: crown-rump length (CRL), nuchal translucency (NT) thickness, ductus venosus pulsatility index (DV-PI), fetal heart rate (FHR), and normal tricuspid valve flow. In addition, blood was sampled from each patient for the determination of the β-subunit of hCG gonadotropin (beta-hCG) and pregnancy-associated plasma protein A (PAPP-A). The determinations were performed using the DELFIA Xpress system (PerkinElmer Life) analyzer. Also, each fetus was examined to determine their hepatic artery flow by measuring the artery’s pulsatility index (HA-PI) and peak systolic velocity (HA-PSV) according to the method described by Zvanca et al. [5]. A transabdominal transducer was used, and the blood flow parameters were measured when the fetus was not moving. In addition, the following conditions were to be met:(1)Image magnification covering the upper torso and the lower chest of the fetus.(2)Longitudinal plane going through the right ventricle of the fetus.(3)Color Doppler showing the inferior vena cava, the ductus venosus and the hepatic artery.(4)Sample volume width–1.0 mm placed in the hepatic artery.(5)Angle of insonation < 30 degrees.(6)Wall filter (WF) set to 120 Hz.(7)Time-axis (sweep speed) 2–3 cm/s.(8)Pulse repetition frequency 2.2–3.3 Hz.

Figure 1 shows an image of a normal hepatic artery flow. 

Subsequently, amniocentesis (AC) was performed on patients with an elevated risk of aneuploidy (cut-off point < 1:300) detected using the FMF (Astraia software) algorithm for karyotype assessment. 

Pregnancy outcome was estimated for each patient. A successful pregnancy outcome was defined as a full-term live birth, i.e., >37 gestational week, with no congenital defects of the neonate and no karyotype abnormalities diagnosed during gestation or after birth, which in the latter case were determined if dysmorphic features were observed during the newborn examination. 

For our study, congenital defects were defined as:(1)Congenital heart defect (CHD), being any of the following: AVSD (atrioventricular septal defect), VSD (ventricular septal defect), CoA (aortic coarctation), TAC (truncus arteriosus communis), HLHS (hypoplastic left heart syndrome), DORV (double outlet right ventricle), ToF (tetralogy of Fallot), PA (pulmonary atresia), TGA (transposition of the great vessels).(2)Another congenital defect, being any of the following: CDH (congenital diaphragmatic hernia), omphalocele, gastroschisis, cleft lip, orofacial cleft, spina bifida, duodenal atresia.

The criteria for including a patient in the adverse pregnancy outcome group were, successively:(1)Pregnancy ended in miscarriage, i.e., before 22 gestational week. In this case, the karyotype was additionally determined in the fetuses in which no such determination had been made during the AC.(2)Pregnancy was terminated due to a specific chromosomal abnormality (trisomy 21, trisomy 18, trisomy 13, Turner syndrome) or congenital defects detected during first- or second-trimester ultrasound examination that provided grounds for termination.(3)Pregnancy ended with intrauterine fetal death (IUFD).(4)Pregnancy ended in preterm labor, i.e., before 37 gestational week.(5)Pregnancy ended in a full-term live birth, i.e., >37 gestational week, with congenital defects or karyotype abnormalities identified.

The results were then analyzed statistically. The non-parametric Mann–Whitney U test and Kruskal–Wallis test were used to calculate the differences between the parameters tested. In addition, correlations were examined using the Spearman’s rank correlation coefficient. In addition, an analysis was performed comprising multiple logistic regression and an area under curve (AUC) calculation. Statistica ver. 13 (StatSoft, Kraków, Poland) software was used for analysis. Approval from the local institutional review board was obtained for the study, and informed consent was obtained from each patient. 

## 3. Results

In the study, 1460 (79.3%) pregnancies ended in a full-term live birth without any known chromosomal or congenital defects (the normal outcome group). Of the 381 patients qualified for the adverse pregnancy outcome group, 187 delivered prematurely (10.1%), 75 miscarried (4%), 74 had their pregnancies terminated (4%), 30 delivered full-term newborns with a diagnosed chromosomal or congenital defect (1.7%), and 15 suffered from intrauterine fetal death (0.9%). 

As shown in Table 1, the fetuses with an adverse pregnancy outcome were shown to have statistically significant lower hepatic arterial pulsatility indexes and higher peak systolic velocities compared to the normal outcome fetuses. Due to the changes we observed, HA-PI values for the 5th percentile in our population (1.19) and HA-PSV values for the 95th percentile in our population (20.11) were derived for the purpose of a thorough logistic regression analysis. These cut-off points were chosen because of a lack of appropriate growth charts for the hepatic artery flow. In addition, with the help of the FMF software, we used the value for the 95th percentile for NT and DV-PI. The logistic regression revealed statistical significance for the predictive model for an adverse pregnancy outcome accounting for: NT > 95pc (OR 2.63 (1.81–3.81)), HA-PI < 5pc (OR 13.99 (4.43–44.23)), HA-PSV > 95pc (OR 11.4 (4.09–31.79)), DV-PI > 95pc (OR 22.47 (9.1–55.36)), maternal age (OR 1.033 (1.007–1.059)), and PAPP-A MoM (OR 0.73 (0.56–0.93)). The AUC for this model was 0.739. In this model, for an FPR of 5% the sensitivity was 45%, the PPV was 70.2%, and the NPV was 86.8%.

Among our patients, karyotype abnormalities were found in 93 fetuses (5%), such as Down syndrome, Edwards syndrome, Patau syndrome or Turner syndrome. The fetuses affected by karyotype abnormalities had statistically significant lower hepatic artery indexes with significantly higher peak systolic velocities (Table 2). The logistic regression demonstrated statistical significance for the predictive model for karyotype abnormalities accounting for: NT > 95pc (OR 6.01 (2.95–12.23)), HA-PSV > 95pc (OR 11.36 (5.51–23.41)), DV-PI > 95pc (OR 20.11 (10.02–40.34)), and PAPP-A MoM (OR 0.26 (0.11–0.59)). The AUC for this model was 0.97. In this model, for an FPR of 5% the sensitivity was 89.2%, the PPV was 48.5%, and the NPV was 99.4%.

Our comparison of the fetuses affected by chromosomal defects with other fetuses included in the adverse pregnancy outcome group also showed statistically significant lower HA-PI and higher HA-PSV values in the former (Table 3).

A closer look at the chromosomal abnormalities and their breakdown into individual defects showed that no significant differences in the Doppler hepatic artery flow assessment existed (Table 4). 

As shown in Table 5, we proved a significantly lower HA-PI in patients who were eligible for pregnancy termination compared to all the other groups. When comparing pregnancy termination patients with the other groups, HA-PSV was significantly higher in the former. In addition, HA-PSV was significantly higher in the miscarriage group compared to the preterm or full-term delivery patients.

Table 6 shows an analysis of hepatic artery flows depending on the presence of a congenital heart defect in the fetus. Fetuses with a congenital heart defect were shown to have a statistically significant lower HA-PI and higher HA-PSV compared to the fetuses without such a diagnosis. The logistic regression demonstrated statistical significance for the predictive model for CHD accounting for: HA-PI < 5pc (OR 7.73 (3.4–17.57)) and DV-PI > 95pc (OR 4.49 (1.95–10.3)). The AUC for this model was 0.75. In this model, for an FPR of 5% the sensitivity was 55.4%, the PPV was 23.1%, and the NPV was 98.5%.

Table 7 and Table 8 demonstrate analyses of the correlations between hepatic flow parameters among the groups under investigation. Statistically significant negative correlations were found between nuchal translucency and the hepatic artery pulsatility index in fetuses born at term, fetuses born preterm, pregnancies ended in miscarriage, fetuses with normal karyotype, and both adverse outcome and normal outcome pregnancies. In addition, we showed positive correlation between NT and peak systolic velocity in the hepatic artery in fetuses born at term, fetuses born preterm, pregnancies ended in miscarriage, terminated pregnancies, fetuses with normal karyotype, and both adverse outcome and normal outcome pregnancies. As for the ductus venosus, statistically significant negative DV-PI correlations with the hepatic artery pulsatility index were found for fetuses born at term, pregnancies ended in intrauterine fetal death, pregnancies ended in miscarriage, terminated pregnancies, fetuses with normal karyotype, and adverse outcome pregnancies. In addition, significantly positive correlations were observed between DV-PI and peak systolic velocity in the hepatic artery for fetuses born at term, pregnancies ended in intrauterine fetal death, fetuses with normal karyotype, and both adverse outcome and normal outcome pregnancies. 

## 4. Discussion

According to our knowledge, this has been the largest (1841 cases) study aiming at evaluating fetal hepatic artery flows. Our main finding was that impaired hepatic artery flows accompanied fetal adverse pregnancy outcomes, aneuploidy, and congenital heart defects. 

The first reports on impaired flows in the hepatic artery were published before the year 2000, where low resistance flows in fetuses with intrauterine growth restriction were described [6]. Other authors have noted that a reduced oxygenated blood flow in the DV contributes to a compensatory increase in the hepatic artery flow aiming at maintaining constant blood flow in the organ (HABR). In their studies, however, this does not only apply to fetuses with intrauterine growth restriction, but also those affected by anemia [7,8]. This implicates that the liver is one of the organs, along with the central nervous system, the heart, and the adrenal glands, that are extremely vital for fetal survival under hypoxic conditions. This effect is probably related to the important hematopoietic function of the fetal liver. Animal studies show that a change in fetal hepatic flow increases its hematopoietic activity [11]. Interestingly, fetuses affected by Down syndrome are prenatally found to show hepatomegaly with an increased hepatic blood flow [12,13,14]. The authors claim that this may cause abnormal hematopoiesis and contribute to acute megakaryoblastic leukemia developing postnatally [15,16]. A similar impaired prenatal blood flow scenario may also apply to fetuses with Edwards syndrome, as postnatally they are much more frequently diagnosed with hepatoblastomas [17,18]. In connection with the above findings, the authors decided a few years ago to look into hepatic artery flows during the first-trimester ultrasound [5,9,10]. One paper shows that fetuses with an adverse pregnancy outcome had a reduced HA-PI and an increased HA-PSV compared to fetuses with normal pregnancy outcomes. However, the paper was based on a relatively small number of patients (*n* = 59) and rather focused on comparing HA flows between fetuses with a normal and an increased NT [10]. In our paper we, too, were able to show that fetuses with an adverse pregnancy outcome had a reduced HA-PI and an increased HA-PSV. In addition, it was these fetuses that we demonstrated to have multiple correlations between hepatic artery flows and the acknowledged adverse pregnancy outcome markers such as NT and DV-PI, as well as successful pregnancy outcome markers such as, say, PAPP-A. In addition, we showed that fetuses with an HA-PI of <5pc are almost 14-times more likely to experience an adverse pregnancy outcome. Fetuses with an HA-PSV >95pc are 11-times more likely to develop such complications.

It should be remembered that for all gynecologists/obstetricians, healthy pregnancy is an extremely important and satisfying part of their daily work. Nevertheless, modern practice requires doctors to reduce the risk of pregnancy complications such as, say, preterm labor or severe cases of early-onset preeclampsia, by ensuring early detection of patients carrying an elevated risk of developing these adverse states. One of the main examples is prevention making use of acetylsalicylic acid in women carrying an increased risk of preeclampsia or using progesterone in women at risk of preterm labor [19,20,21,22]. Therefore, researchers have been attempting to identify other markers causing the various forms of complications during pregnancy. The flow in the ductus venosus is an example of such a marker. An increased DV-PI, as well as the presence of a reversed a-wave in the DV, correlate significantly with pregnancy complications such as aneuploidy, miscarriage, and intrauterine fetal death [23,24,25]. 

Our results also showed that DV-PI is higher in fetuses with an adverse pregnancy outcome or those with aneuploidy. We additionally proved that a DV-PI > 95pc increases the risk of an adverse pregnancy outcome more than 22 times and the risk of aneuploidy more than 20 times.

The search for new markers for chromosomal aberrations continues, which may result in an improved detection rate of the aforementioned types of aneuploidy in the first trimester of pregnancy. So far, the evaluation proposed in 2008, which achieves a detection rate of 91% at a false-positive rate (FPR) of 5% for trisomy 21, continues to be the most effective [2]. The addition of other markers, such as the presence of the nasal bone, or tricuspid insufficiency, has been shown to improve the detection rate of aneuploidy, reducing the FPR at the same time [26,27,28,29]. Fetuses with karyotype abnormalities were reported to have a high HA-PSV and a low HA-PI. In addition, significant negative correlations were proved to exist between the hepatic artery flow and the DV flow, and positive correlations were shown to exist between it and NT [5,9,10]. In addition, two of these papers show that a higher HA-PSV and a lower HA-PI are not only found in fetuses with chromosomal abnormalities but also in fetuses in which the risk of this aneuploidy is increased, even though their karyotype is normal [5,9]. The main focus of those papers was on trisomy 21 with only a handful of other chromosomal abnormalities examined, which was an insufficient basis to assess the usefulness of hepatic artery flow measurements in such cases. As we have shown in our study, as well, HA-PI is lower and HA-PSV is higher in fetuses with confirmed chromosomal aberrations compared to fetuses with a normal karyotype. Our more detailed analysis allowed us to show that fetuses with an HA-PSV >95pc are 11 times more likely to have aneuploidy, while the AUC for our model was 0.97. When studying correlations in aneuploid fetuses, we were not able to show any significant relationships between hepatic artery flow and DV or NT. This may indicate that the flow in the hepatic artery may be an independent additional marker for these aberrations. In addition, we examined the differences in hepatic artery flows depending on the type of the chromosomal abnormality present. In this respect, we were not able to show any differences between the groups, which may mean that the presence of aneuploidy alone disturbs the normal hepatic artery flow.

Early detection of CHD is another important aspect of prenatal testing. Currently, the prenatal CHD detection rate is estimated to be approx. 60% [30,31]. So far, NT and the DV flow have been relatively well-researched as markers for the risk of developing CHD. Many authors have shown that first-trimester fetuses diagnosed with an increased NT, an elevated DV pulsatility index, or the presence of an a-wave in the DV, carry a higher risk of congenital heart defects, regardless of the risk of aneuploidy [32,33,34]. In the case of these fetuses, it is important that second-trimester echocardiography is performed as a follow-up. Nevertheless, as noted above, no satisfactory CHD detection rate has been reached as yet. The flow in the hepatic artery is an additional parameter that could guide us towards obtaining a better insight into the fetus for a risk of CHD. We have been able to show that these cases of fetuses have significantly lower pulsatility index values, accompanied by significantly higher peak systolic velocities in the HA. A more thorough study of these parameters helped us note that a DV-PI > 95pc increases the risk of CHD almost 4.5 times, while an HA-PI < 5pc increases that risk more than 7 times. The AUC for this model was 0.75. However, with an assumed FPR of 5%, the sensitivity of this model was only 55.4%. Nevertheless, it should be stressed that the focus here is on the first trimester, while most CHD cases are detected much later. 

In view of the above, it appears appropriate that a discussion should be started on hepatic artery flow assessments in the first trimester of pregnancy. We are aware that that beginner sonographers may find it difficult to identify the signal of the hepatic artery, especially where it indicates normal flow parameters. The fetal hepatic artery is in close proximity to the ductus venosus, which is why when imaging the ductus venosus a strong signal that corresponds to the hepatic artery is frequently observed. For persons trained in fetal ultrasound imaging, however, this assessment should not pose more difficulty than making the routine first-trimester parameter evaluations.

One of the weaknesses of our study was that no karyotype study was carried out in children born at term without dysmorphic features as this might have led to somewhat different results, but the cost of such an approach would have been unacceptable for us. In addition, due to an initial lack of reporting, we do not have accurate data about the numbers of patients excluded from the study. Certainly, if we wanted to extrapolate the detection rate of an adverse pregnancy outcome to the general population, we should also investigate patients with pre-pregnancy diseases or with a history of complications in their previous pregnancies, who according to our initial assumptions did not qualify for inclusion in the present study. 

## 5. Conclusions

Expanding the first-trimester screening by the addition of the hepatic artery flow assessment may contribute to improving the detection rate of fetuses carrying the risk of developing adverse pregnancy outcomes. In particular cases, this could mean an ability to detect chromosomal abnormalities or congenital heart defects. However, multicenter studies would be needed to confirm our observations.

## Figures and Tables

**Figure 1 jcm-11-02095-f001:**
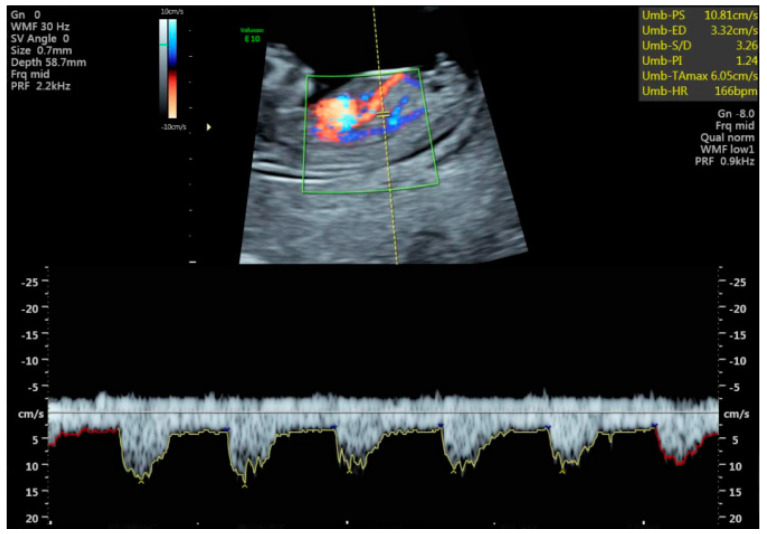
Longitudinal plane view of 12-week fetus showing the umbilical vein, the ductus venosus and the descending thoracic aorta on color flow. The hepatic artery is the vessel coming into close contact with the ductus venosus.

**Table 1 jcm-11-02095-t001:** Differences between selected first-trimester screening parameters depending on the adverse pregnancy outcome.

	Adverse Pregnancy Outcome		
	Yes (*n* = 381)	No (*n* = 1460)	*p*
	Median (Min–Max)	Median (Min–Max)	
FHR	161.00 (138.00–191.00)	160.00 (143.00–197.00)	0.017
NT	2.10 (1.10–13.00)	1.75 (0.80–10.50)	<0.001
HA-PI	1.43 (0.11–2.70)	1.47 (1.01–2.70)	<0.001
HA-PSV	12.27 (7.12–43.00)	11.45 (5.41–26.30)	<0.001
DVPI	1.20 (0.56–2.43)	1.10 (0.61–9.92)	<0.001
beta-HCG MoM	1.01 (0.04–7.26)	1.04 (0.17–7.68)	0.56
PAPP-A MoM	0.68 (0.05–3.54)	0.94 (0.15–4.32)	<0.001
	Mean ± SD	Mean ± SD	
Age	31.77 ± 5.97	30.5 ± 5.25	<0.001
Weight	65.05 ± 10.46	65.21 ± 7.9	0.006

**Table 2 jcm-11-02095-t002:** Differences between selected first-trimester screening parameters depending on the karyotype.

	Karyotype		
	Abnormal (*n* = 93)	Normal (*n* = 1741)	*p*
	Median (Min–Max)	Median (Min–Max)	
FHR	159.00 (142.00–182.00)	160.00 (138.00–197.00)	0.93
NT	4.00 (1.40–12.00)	1.80 (0.80–13.00)	<0.001
HA-PI	0.84 (0.54–2.53)	1.47 (0.11–2.70)	<0.001
HA-PSV	20.90 (7.79–43.00)	11.55 (5.41–28.70)	<0.001
DVPI	1.65 (0.78–2.43)	1.10 (0.56–9.92)	<0.001
beta-HCG MoM	1.12 (0.12–4.37)	1.03 (0.04–7.68)	0.85
PAPP-A MoM	0.44 (0.09–2.09)	0.92 (0.05–4.32)	<0.001
	Mean ± SD	Mean ± SD	
Age	32.09 ± 6.69	30.69 ± 5.35	0.14
Weight	65.15 ± 8.36	65.64 ± 10.83	0.65

**Table 3 jcm-11-02095-t003:** Differences between selected first-trimester screening parameters in adverse pregnancy outcome pregnancies depending on the karyotype.

	Karyotype		
	Abnormal (*n* = 93)	Normal (*n* = 288)	*p*
	Median (Min–Max)	Median (Min–Max)	
FHR	159.00 (142.00–182.00)	161.00 (138.00–191.00)	0.44
NT	4.00 (1.40–12.00)	1.90 (1.10–13.00)	<0.001
HA-PI	0.84 (0.54–2.53)	1.47 (0.11–2.70)	<0.001
HA-PSV	20.90 (7.79–43.00)	11.90 (7.12–28.70)	<0.001
DVPI	1.65 (0.78–2.43)	1.11 (0.56–2.10)	<0.001
beta-HCG MoM	1.12 (0.12–4.37)	1.01 (0.04–7.26)	0.91
PAPP-A MoM	0.44 (0.09–2.09)	0.80 (0.05–3.54)	<0.001
	Mean ± SD	Mean ± SD	
Age	32.09 ± 6.69	31.67 ± 5.72	0.96
Weight	65.15 ± 8.36	64.86 ± 10.36	0.43

**Table 4 jcm-11-02095-t004:** Differences between selected first-trimester screening parameters depending on the type of aneuploidy.

	Karyotype Defect				
	T13 † (*n* = 10)	T18 †† (*n* = 17)	T21 ††† (*n* = 57)	Turner Syndrome †††† (*n* = 9)	*p*
	Median (Min–Max)	Median (Min–Max)	Median (Min–Max)	Median (Min–Max)	
FHR	172.5 (142–182)	158 (150–169)	157 (142–182)	178 (160–180)	<0.001
NT	5.55 (2.00–8.00)	4.40 (1.70–9.60)	3.40 (1.40–9.50)	8.90 (5.80–12.00)	<0.001
HA-PI	1.04 (0.75–1.99)	0.85 (0.68–1.90)	0.82 (0.54–2.53)	0.92 (0.76–1.45)	0.098
HA-PSV	20.90 (8.57–23.10)	23.20 (19.70–43.00)	20.70 (7.79–42.20)	20.40 (10.66–33.20)	0.066
DVPI	1.56 (1.17–1.99)	1.65 (1.08–2.40)	1.65 (0.78–2.43)	1.24 (0.99–1.92)	0.3
beta-HCG MoM	0.51 (0.38–3.45)	0.45 (0.12–1.92)	1.30 (0.33–4.37)	1.50 (0.24–2.65)	<0.001
PAPP-A MoM	0.28 (0.22–1.45)	0.32 (0.09–0.73)	0.58 (0.15–2.09)	0.46 (0.22–1.52)	0.0012
	Mean ± SD	Mean ± SD	Mean ± SD	Mean ± SD	
Age	29.2 ± 6.86	29.64 ± 5.07	33.43 ± 7.08	31.44 ± 4.74	0.11
Weight	64.9 ± 8.06	67 ± 11.44	66.01 ± 11.59	61.55 ± 7.09	0.64

Post-hoc: **FHR**: † vs. ††† *p* = 0.04, †† vs. †††† *p* = 0.01, ††† vs. †††† *p* = <0.001; **NT**: ††† vs. †††† *p* = <0.001; **beta-HCG**: † vs. ††† *p* = 0.01 †† vs. ††† *p* = <0.001; **PAPP-A**: † vs. ††† *p* = 0.01 †† vs. ††† *p* = 0.01.

**Table 5 jcm-11-02095-t005:** Differences between selected first-trimester screening parameters depending on pregnancy outcome.

	Pregnancy Outcome					
	Labor at Term † (*n* = 1483)	IUFD †† (*n* = 15)	Miscarriage ††† (*n* = 75)	Terminated Pregnancy †††† (*n* = 74)	Preterm Labor ††††† (*n* = 187)	*p*
	Median (Min–Max)	Median (Min–Max)	Median (Min–Max)	Median (Min–Max)	Median (Min–Max)	
FHR	160 (143–197)	160 (146–177)	162 (144–190)	162 (138–191)	161 (142–189)	0.053
NT	1.80 (0.80–10.50)	1.80 (1.10–9.50)	2.00 (1.20–12.00)	4.25 (1.10–13.00)	1.90 (1.10–10.80)	<0.001
HA-PI	1.46 (0.54–2.70)	1.52 (0.72–2.66)	1.43 (0.11–2.69)	1.11 (0.68–2.58)	1.46 (0.66–2.70)	<0.001
HA-PSV	11.50 (5.41–33.20)	11.30 (7.43–41.20)	12.85 (7.54–28.70)	20.11 (7.79–42.20)	11.74 (7.12–43.00)	<0.001
DVPI	1.10 (0.61–9.92)	1.20 (0.80–1.86)	1.30 (0.70–2.43)	1.51 (0.87–2.40)	1.10 (0.56–1.95)	<0.001
beta-HCG MoM	1.04 (0.17–7.68)	0.93 (0.14–3.10)	1.09 (0.04–7.26)	1.17 (0.12–6.07)	0.99 (0.14–5.73)	0.76
PAPP-A MoM	0.94 (0.15–4.32)	0.73 (0.05–1.90)	0.59 (0.05–3.54)	0.41 (0.09–3.03)	0.84 (0.12–3.41)	<0.001
	Mean ± SD	Mean ± SD	Mean ± SD	Mean ± SD	Mean ± SD	
Age	30.59 ± 5.32	29.33 ± 6.62	33.46 ± 5.11	31.41 ± 5.88	30.84 ± 5.83	<0.001
Weight	65.21 ± 7.98	62.46 ± 10.81	64.74 ± 9.08	64.7 ± 9.88	65.51 ± 11.1	0.049

Post-hoc: **NT** † vs. ††† *p* < 0.001 † vs. †††† *p* < 0.001 † vs. ††††† *p* = 0.002; †† vs. †††† *p* = 0.006; ††† vs. †††† *p* < 0.001 ††† vs. ††††† *p* = 0.02; †††† vs. ††††† *p* < 0.001; **HA-PI** † vs. †††† < 0.001; †† vs. †††† *p* = 0.02; ††† vs. †††† *p* = 0.002; †††† vs††††† *p* < 0.001; **HA-PSV** † vs. ††† *p* < 0.001 † vs. †††† *p* < 0.001; †† vs. †††† *p* = 0.02; ††† vs. †††† *p* = 0.02 ††† vs. ††††† *p* = 0.03; †††† vs. ††††† *p* < 0.001; **DVPI** † vs. ††† *p* < 0.001 † vs. †††† *p* < 0.001; ††† vs. ††††† *p* < 0.001; †††† vs. ††††† *p* < 0.001; **PAPP-A** † vs. ††† *p* < 0.001 † vs. †††† *p* < 0.001; ††† vs. ††††† *p* = 0.002; †††† vs. ††††† *p* < 0.001; **Age** † vs. ††† *p* < 0.001; ††† vs.††††† *p* = 0.002.

**Table 6 jcm-11-02095-t006:** Differences between selected first-trimester screening parameters depending on present or absent fetal CHD diagnosis.

	CHD		
	Yes (*n* = 56)	No (*n* = 1785)	*p*
	Median (Min–Max)	Median (Min–Max)	
FHR	160.00 (142.00–180.00)	160.00 (138.00–197.00)	0.68
NT	2.35 (1.10–12.00)	1.80 (0.80–13.00)	<0.001
HA-PI	1.30 (0.54–2.65)	1.46 (0.11–2.70)	<0.001
HA-PSV	13.82 (7.43–43.00)	11.65 (5.41–42.20)	<0.001
DVPI	1.30 (0.78–2.11)	1.10 (0.56–9.92)	<0.001
beta-HCG MoM	0.89 (0.12–4.44)	1.04 (0.04–7.68)	0.08
PAPP-A MoM	0.63 (0.10–2.24)	0.91 (0.05–4.32)	<0.001
	Mean ± SD	Mean ± SD	
Age	32.01 ± 6.84	30.72 ± 5.37	0.18
Weight	64.66 ± 9.91	65.19 ± 8.45	0.24

**Table 7 jcm-11-02095-t007:** Correlations between hepatic artery flows and selected first-trimester screening parameters depending on the pregnancy outcome (ns = not significant).

		Labor at Term	IUFD	Miscarriage	Termination of Pregnancy	Preterm Labor
FHR	HA-PI	*p* < 0.001 R = 0.31	*p* < 0.001 R = 0.73	ns	ns	*p* < 0.001 R = 0.36
	HA-PSV	*p* < 0.001 R = −0.33	ns	ns	ns	*p* < 0.001 R = −0.3
NT	HA-PI	*p* < 0.001 R = −0.55	ns	*p* < 0.01 R = −0.29	ns	*p* < 0.001 R = −0.5
	HA-PSV	*p* < 0.001 R = 0.57	ns	*p* < 0.001 R = 0.61	*p* < 0.001 R = 0.38	*p* < 0.001 R = 0.56
DVPI	HA-PI	*p* < 0.02 R = −0.05	*p* < 0.02 R = −0.58	*p* < 0.03 R = −0.24	*p* < 0.002 R = −0.34	ns
	HA-PSV	*p* < 0.01 R = 0.06	*p* < 0.03 R = 0.54	ns	ns	ns
beta-HCG MoM	HA-PI	ns	ns	ns	ns	ns
	HA-PSV	ns	ns	ns	ns	ns
PAPP-A MoM	HA-PI	ns	ns	ns	ns	ns
	HA-PSV	ns	ns	ns	ns	ns

**Table 8 jcm-11-02095-t008:** Correlations between hepatic artery flows and selected first-trimester screening parameters depending on the karyotype and the adverse pregnancy outcome (ns = not significant).

		Abnormal Karyotype	Normal Karyotype	Adverse Outcome	Normal Outcome
FHR	HA-PI	*p* < 0.01 R = 0.25	*p* < 0.001 R = 0.31	*p* < 0.001 R = 0.21	*p* < 0.001 R = 0.33
	HA-PSV	ns	*p* < 0.001 R = −0.3	*p* < 0.01 R = −0.12	*p* < 0.001 R = −0.34
NT	HA-PI	ns	*p* < 0.001 R = −0.51	*p* < 0.001 R = −0.47	*p* < 0.001 R = −0.54
	HA-PSV	ns	*p* < 0.001 R = 0.58	*p* < 0.001 R = 0.62	*p* < 0.001 R = 0.56
DVPI	HA-PI	ns	*p* < 0.02 R = −0.05	*p* < 0.001 R = −0.32	ns
	HA-PSV	ns	*p* < 0.001 R = 0.07	*p* < 0.001 R = 0.26	*p* < 0.03 R = 0.05
beta-HCG MoM	HA-PI	ns	ns	ns	ns
	HA-PSV	ns	ns	ns	ns
PAPP-A MoM	HA-PI	ns	ns	*p* < 0.001 R = 0.17	ns
	HA-PSV	ns	ns	*p* < 0.001 R = −0.17	ns

## Data Availability

The data presented in this study is available upon request from the author for correspondence. The data is not publicly available, as not all patients agreed to publicly disclose the data.

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
