# Peer review of "First-Trimester Fetal Hepatic Artery Examination for Adverse Outcome Prediction"

_jcm, 2022, doi:10.3390/jcm11082095_

Round 1
Reviewer 1 Report
This article demonstrates that the use of first trimester hepatic arterial flow assessment can contribute to other first trimester screening markers to predict adverse pregnancy outcome, including chromosomal abnormalities, pregnancy loss and some fetal anomalies.
Overall this is a well conducted study which contributes valuable information to the field of prenatal diagnosis.
The introduction provides a nice summary of the literature, although most of the reference quoted are recent, it would be nice to see some more recent literature replacing some of the older (reference 6).
The patients and methods section descibes a large study. The methods are well described and are reproducable. Pregnancy outcomes were defined although I find that some of the congenital defects lack relevence. Placing cleft lip into the same category as CDH degrades the relevence. I am not sure why these particular congenital defects were chosen from a myriad of others, or whether this is just a sample. Perhaps it could be explained why these ones and not others (i.e. urogenital defects, sacrococcygeal teratoma, etc. were chosen). The abnormal pregnancy ourcomes were well described.
The tables and narrative in the result section provides the reader with a good summary of the findings.
The discussion provides a good summary of the merits of the findings and their relevance. I would like to see a little more discusion of the weaknesses of the paper and discussion as to whether this impacted the findings.
The concusion is consistent with the evidence and arguments provided.
Overall this manuscript is a relevent and the work helps to contribute to the advancement of prenatal diagnosis. References are appropriate. Unfortunately although the manuscript is well structured, grammatical and spelling errors throughout are distracting. I would suggest that the authors have the manuscript reviewed by an English colleague.
Reviewer 2 Report
The paper aimed at showing the correlation between changes in fetal hepatic artery flow and pregnancy adverse outcomes. I believe the subject of prenatal diagnosis is ever evolving and any insight into upgrading our abilities is greatly welcome.
I enjoyed reading the paper very much. I think parts of it are well written.
Introduction
Well written and summarizes the main focus of the study. Except for minor spelling issues, it requires no further editing.
Methodology
1. You mention the number 1841 patients that were recruited. How many patients did you try to recruit in total? How many refused? Were there other considerations for patient recruitment?
2. You did not mention any exclusion criteria. Were there none? Were all women eligible?
3. Though the description of the US procedure is thorough, I think it would benefit the readers to see a figure of the US itself and the measurements (especially since this is not a routine test and you wish to prove the applicability of the test for common use in prenatal diagnosis)?
4. What were the references for low/high HA-PI/PSV? How did you determine the cutoffs? IS that based on references 6-7. It would be beneficial to readers to understand the normograms/cutoffs of the flow in the fetal hepatic artery.
5. It is unclear what the primary outcomes of the study is and what the secondary outcomes are? This is basic methodology.
6. Since no primary outcome was defined, I understand calculating a sample size as one would do in a prospective study was irrelevant? This is also a major flaw if not done.
Results
7. Age and Weight are continuous parameters and should be displayed as means and standard deviations.
8. It is not mentioned in any place what the rate of maternal comorbidities was. Obviously, pre-gestational diabetes and other afflictions may be associated with adverse pregnancy outcomes, as well as chronic hypertension and other chronic conditions. The same is true for pregnancy complications that may also be associated with preterm delivery and additional adverse outcomes. This information must be added and correlations must be done.
9. The study is lacking correlations between the results of the hepatic artery doppler measurements and known associated first trimesters tests such as NT. Why not do those correlations. They are much more interesting than the correlations between the measurements and the outcomes.
10. You are missing basic test parameter calculation including sensitivity, specificity, negative and positive predictive values. There are much more important in validating your results than the calculations you did. You missed the point, I think. I would also compare the test parameters of the hepatic artery dopplers to the test parameters of the other tests you evaluated including the NT.
Discussion
11. It is accustomed that the discussion usually begins with the main findings of the study and then go on to compare the findings with previous publications. Your first paragraph (lines 190-198) describes that you conducted the largest study, the proximity of the hepatic artery to the ductus venosus and the ability of beginner sonographers to attain it`s measurements. I find none of this relevant in the beginning of a discussion, maybe more towards the end if at all.
12. The next 2 paragraphs (lines 199-227) discuss the DV flow and it is only at the end of that 2nd paragraph that you refer to your results of the correlation between DV flow changes and adverse outcomes in pregnancy. I do not understand the purpose of these 2 paragraphs in establishing the results you found and the subject of changes in the hepatic artery flow.
13. In the next paragraph (lines 228-231) you mentioned finding a similar correlation between hepatic artery flow changes and adverse pregnancy outcomes just as references 5,9,10, except that your sample size was larger (the 2nd time you mentioned this). There is no comparison between what they studied (in terms of outcomes) and what you study found in terms of similarities and differences.
14. I do not understand the statement in lines 243-244 regarding the fact that "no significant correlations were 243 found with the DV or NT in aneuploid fetuses". This does not correlate with the numbers you reported in Table 4 and previous reports that have found significant correlations between high NTs and aneuploid fetuses which you did not mention here. This is a grave mistake.
15. In the last paragraph of the discussion (lines 246-257), again you give a long intro regarding the association between CHD and first trimester markers but only mention your results in the last 2 lines.
16. While this discussion is very intellectual, it does not focus in any way on the study results, does not strengthen your case and at most, elaborates on 1st trimester screening in general. I think the discussion is ill-written, way off point and has nothing to do with your study.
17. With all the faults of the current manuscript, I think that the conclusion is quite pretentious and does not reflect proper scientific evidence.
Round 2
Reviewer 2 Report
The revisions are satisfactory.
There are some fundamental issues with the methodology and discussion that the authors should try to improve in future studies.
All in all, the subject is very important and the results may serve as a base for future studies.